

# A comparison of clustering methods for biogeography with fossil datasets

Matthew J. Vavrek

Department of Natural History, Royal Ontario Museum, Toronto, Ontario, Canada

## ABSTRACT

Cluster analysis is one of the most commonly used methods in palaeoecological studies, particularly in studies investigating biogeographic patterns. Although a number of different clustering methods are widely used, the approach and underlying assumptions of many of these methods are quite different. For example, methods may be hierarchical or non-hierarchical in their approaches, and may use Euclidean distance or non-Euclidean indices to cluster the data. In order to assess the effectiveness of the different clustering methods as compared to one another, a simulation was designed that could assess each method over a range of both cluster distinctiveness and sampling intensity. Additionally, a non-hierarchical, non-Euclidean, iterative clustering method implemented in the R Statistical Language is described. This method, Non-Euclidean Relational Clustering (NERC), creates distinct clusters by dividing the data set in order to maximize the average similarity within each cluster, identifying clusters in which each data point is on average more similar to those within its own group than to those in any other group. While all the methods performed well with clearly differentiated and well-sampled datasets, when data are less than ideal the linkage methods perform poorly compared to non-Euclidean based $k$-means and the NERC method. Based on this analysis, Unweighted Pair Group Method with Arithmetic Mean and neighbor joining methods are less reliable with incomplete datasets like those found in palaeobiological analyses, and the $k$-means and NERC methods should be used in their place.

## INTRODUCTION

Clustering, defined as "a classificatory method which optimizes intra-group homogeneity" (*Lance & Williams, 1967*), is one of the most frequently used forms of multivariate analysis in palaeoecology (*Hammer, Harper & Ryan, 2001*). One of the areas in which cluster analysis is commonly used is studying patterns of biogeography amongst species assemblages. Cluster analysis has been used in palaeoecological studies on groups as diverse as vertebrates (*Shubin & Sues, 1991*; *Holtz, Chapman & Lamanna, 2004*; *Fröbisch, 2009*; *Gates et al., 2010*; *Noto & Grossman, 2010*; *Donohue, Wilson & Breithaupt, 2013*), invertebrates (*Schwimmer, 1975*; *Clapham & James, 2008*), foraminifera (*Collins, 1993*) and plants (*LePage et al., 2003*), and assemblages spanning the Ediacaran (*Clapham, Narbonne & Gehling, 2003*) to the Pleistocene (*Wolfe, 2000*). With the rise of

Corresponding author
Matthew J. Vavrek,
matthew@matthewvavrek.com

large datasets of fossil species occurrences (e.g. Paleobiology Database, MioMAP (*Carrasco et al., 2005*), FAUNMAP (*Graham & Lundelius, 2010*), NOW (*Fortelius, 2015*); see *Uhen et al., 2013* for recent review) with hundreds or thousands of records, semi-automated methods such as clustering are becoming more and more necessary to find underlying patterns in these highly complex collections. As the use of cluster analysis in palaeobiology has steadily expanded, so too have the types of methods used. Although the underlying purpose of these methods is the same (i.e. to delimit different groups from one another), their approaches and assumptions are often quite different. For example, some cluster analysis methods (e.g. Unweighted Pair Group Method with Arithmetic Mean/UPGMA, neighbour–joining) use a hierarchical approach to grouping data (*James & McCulloch, 1990*; *Shi, 1993*).

Other common methods include partitioning techniques, such as *c*-means or *k*-means, which may try to optimize groups by minimizing relative distances based on a chosen index (*Hartigan & Wong, 1979*). Although clustering methods may be widely used, their effectiveness relative to one another is less well known, in particular with the often sparse datasets used in palaeobiological studies. In order to examine the relative efficacy of these different clustering methods with species occurrence data, a dataset where the "true" clustering relationship is known is required. To generate multiple simulated datasets with established clustering relationships, I created an R function which could create a species occurrence database that could then be used to test the efficiency of the methods over a large number of trials.

In addition to the analysis of the various clustering methods commonly used, I also describe here an R function for a non-Euclidean, non-hierarchical clustering method termed here Non-Euclidean Relational Clustering (NERC), an iterative method that uses agglomerative clustering with post-clustering optimization. The efficacy of this function is tested in comparison to the more traditional methods.

## MATERIALS AND METHODS

### The NERC function

The algorithm's execution can be broken down into three distinct steps (after *Lance & Williams, 1967*): the initialization of clusters; the allocation of new elements to a cluster; and finally an iterative reallocation process whereby the clusters are optimized. The first step, initialization of the clusters, begins by sampling a number of elements equal to the requested number of final clusters. Each of these selected samples is assigned randomly to a different initial cluster. In the second step, the function searches for the greatest similarity (smallest value in a dissimilarity matrix) between any unassigned sample and any assigned sample. The unassigned sample with the highest similarity is assigned to the same group as that which it shares the greatest similarity, similar to Single Linkage Clustering Analysis (*Gower & Ross, 1969*). This process then repeats, until all samples are assigned to a cluster. At the end of the second step, if any group has only one member the process restarts from the first step.

As a final step, an optimization of the clusters is performed. To begin, each individual sample within the entire set is assessed for its average similarity to every cluster.

The similarity is based on the average pairwise distance from a sample to every member of a cluster (excluding the sample itself in the case of the cluster it had been assigned to). If a sample has a greater similarity to another cluster other than the one it has been assigned to, the optimization routine will reassign the sample to the cluster that it had the most similarity to. If more than one sample is in a suboptimal cluster, only one sample, chosen at random, will be reassigned at a time. After a sample has been reassigned, the average pairwise distances will be calculated again before another sample is reassigned (if necessary). If all the samples are in the cluster with which they have the greatest average similarity then the cycle is complete. At present, an upper limit of 1000 reassignments has been set so as to avoid an infinite loop if there is no solution where every sample is in its optimal grouping. The process will find a local, but not necessarily global, optimum by minimizing the overall dissimilarity within clusters. Because the method is heuristic in nature, it is best to repeat the clustering process many times.

## Implementation of NERC

The R Statistical Language (*R Development Core Team, 2015*) was used to implement the NERC function. The R Language is cross platform, Open Source and free to use, and is widely used in statistical research, making it easy to extend with new functions and packages. The package fossil (*Vavrek, 2011*) with all of the functions discussed in this paper is available through the Comprehensive R Archive Network (CRAN) at http://cran.r-project.org/web/packages/fossil/. All data analysis and figure creation was done using R v3.2.1 on a Mac OS X 10.10 system. For a full copy of the R code used in the calculations and figures, please consult the Supplementary Materials.

The R implementation of the NERC function has one required and three optional arguments, and takes the form:

rclust (dist, clusters = 2, rand = 1000, counter = FALSE)

The only required argument is a distance or dissimilarity matrix (the dist argument), either as a full matrix or lower triangle. The first optional argument (clusters) is the number of groups to be created. The number of groups used must be a positive integer equal to or greater than 2 but no greater than 1/2 the total number of samples. The minimum value represents the smallest number of clusters without placing all samples within one group, and the maximum value prevents clusters of one. The default value for the number of clusters is set to 2. The second optional argument gives the number of times the clustering process should be run. Because the method should be run many times to have a better chance of finding the global optimal solution, this option has a default value of 1000. The last optional argument (counter) specifies whether to print the current run. Note that at this point the R function returns only the result with the smallest average within group distances overall.

## Data simulation and comparisons

In order to test the efficacy of NERC in comparison to several other cluster methods, I also created a simple function to simulate a species abundance data set. This

function, called sim.occ(), creates a matrix of sites (columns) and species (rows) with a known clustering solution. The number of species, localities, regions (clusters), sample size and proportion of regional endemicity can all be adjusted. Each specific 'region' in the simulated set contains a number of 'cosmopolitan' species that are found in every region, as well as 'endemic' species that are found in only that particular region. To obtain a sample for a single locality, a randomized log-normal distribution is applied to the total possible species pool for a given region; the parameters are set so that any given locality will have several abundant species, a large number of less common species, and some species which are not present. A log-normal distribution was used as it is one of the most common species abundance distributions found in empirical samples of modern habitats (*Preston, 1962*; *Gaston & Blackburn, 2000*; *Magurran, 2004*). For every sample, a new randomized log-normal distribution was created from the parent region species pool. The average number of specimens can be varied to simulate different sampling intensities. The full R code for the function can be found within the fossil package.

The simulated data was clustered using 6 different combinations of methods and input matrices: single linkage, complete linkage, UPGMA, *k*-means on a db-RDA ordination using both Euclidean and a non-Euclidean distance measure, and NERC. For those methods, that provide hierarchical clusters, discrete clusters were made using the cutree() function. The db-RDA ordination was performed using the capscale function in the vegan (*Oksanen et al., 2015*) package.

Most functions used require a distance matrix as input, rather than raw species values. In order to convert the occurrence matrices to dissimilarity matrices, the ecol.dist() function was used, with the Sørensen (sometimes called Dice) dissimilarity index used to calculate pairwise dissimilarities. The Sørensen dissimilarity index was used because it is one of the most commonly used indicices, and is regarded as one of the most effective presence/absence dissimilarity measures (*Southwood & Henderson, 2000*; *Magurran, 2004*). Although the sim.occ() function did create abundance-based occurrence matrices, the Sørensen dissimilarity index is presence/absence based, in effect converting the data. Although discarding abundance data is not generally recommended in actual analyses, presence/absence data is typically more common in palaeontological datasets, so using the Sørensen dissimilarity index created a more realistic scenario.

The six methods were tested to see how well they performed both with varying levels of endemicity (or differentiation between clusters; Fig. 1) as well as with varying levels of sampling intensity. A simulated occurrence matrix was created 1000 times for each level of differentiation or sampling intensity, and then clustered to obtain averaged performance values for all five clustering methods. Each of the simulations consisted of 30 samples from 3 different endemic regions, for a total of 90 samples to be used in the cluster analysis. Because of the parallel nature of this simulation, the foreach (*Revolution Analytics and Steve Weston, 2015b*), and doMC (*Revolution Analytics and Steve Weston, 2015a*) parallel computing packages for R were also used. The visualization of cluster distinctiveness in Fig. 1 was created using the NMDS function provided by the ecodist package (*Goslee & Urban, 2007*).

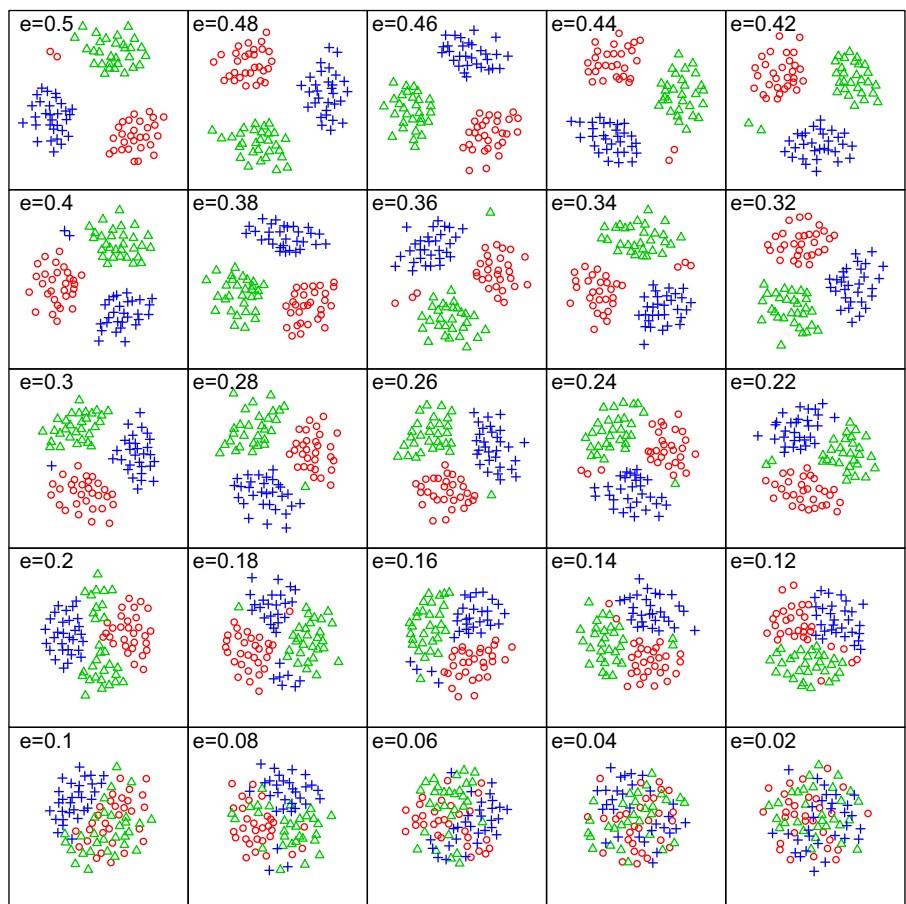

**Figure 1 Variation in group distinctiveness for simulated data.** Visualization of the changing endemicity of clusters (i.e. distinctiveness) and how it alters the clustering of sites in an NMDS plot for the simulated biogeographic data sets. 'e' is the proportion of all species that are endemic to only one biogeographic region. A higher proportion of endemics results in more distinctive clusters, while a lower proportion of endemics results in less distinctive clusters.

For the simulated biogeographic datasets, the "true" clustering was known, and so the results of each clustering method could be compared to this *a priori* grouping. The Rand Index (*Rand, 1971*; *Hubert & Arabie, 1985*) is method to compare two clustering outcomes and calculates an index of similarity, with a value of 1 being a perfect match. The original formula for this index, however, had a lower bound that fluctuated, depending on group sizes and numbers (*Hubert & Arabie, 1985*). A modification of this original formula, given by *Hubert & Arabie (1985)*, scaled the value so that the greatest mathematically possible difference would always be 0, with the upper bound still set to 1. This modification is referred to as the Adjusted Rand Index (ARI). In the fossil package, both functions are provided, although only the ARI is used to calculate the effectiveness of the clustering methods in this paper.

## RESULTS

Overall, the NERC and non-Euclidean *k*-means methods were the most effective at recovering the original groupings across the different levels of regional endemicity (Fig. 2),

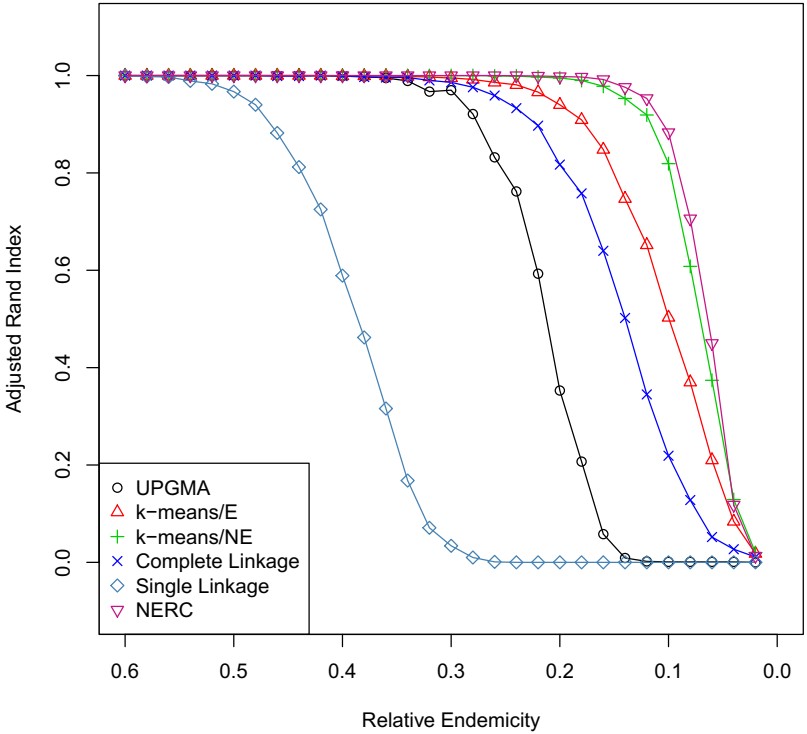

**Figure 2 Comparison of cluster methods with varying group distinctiveness.** Response of various clustering methods to the distinctiveness of clusters as given by the proportion of endemics (i.e. a higher endemicity creates more highly differentiated clusters). The values for each method at any given level of endemicity is the average Adjusted Rand Index comparing the known solution and the calculated solution over 1000 simulations.

with the NERC slightly outperforming the non-Euclidean $k$-means. Using a Euclidean distance metric for the $k$-means method, even when the rest of the method and dataset are kept the same, led to a notable reduction in performance. Complete linkage and UPGMA were readily able to recover the correct clusters when the groups were relatively distinct. However, when the simulated clusters were less distinct their effectiveness quickly declined. Single linkage clustering was least effective and, produced unreliable results even at levels where all the other methods easily found the proper clustering arrangement.

For the differing levels of sampling intensity (Fig. 3), the NERC method and non-Euclidean $k$-means methods were again the most effective at recovering an accurate signal, although in this instance the $k$-means was slightly more effective. Overall, complete linkage and UPGMA gave accurate results when sampling intensity was high, but their performance was very poor with sparsely sampled data. Single linkage was again the least effective of all the methods tested.

## DISCUSSION

All cluster methods performed well when clusters were very distinct and sampling intensity was high. However, in cases where biogeographic clusters were less distinct or sampling was poor, the db-RDA/$k$-means and the NERC methods were best able to recover the original clusters compared to the other tested clustering methods. Among

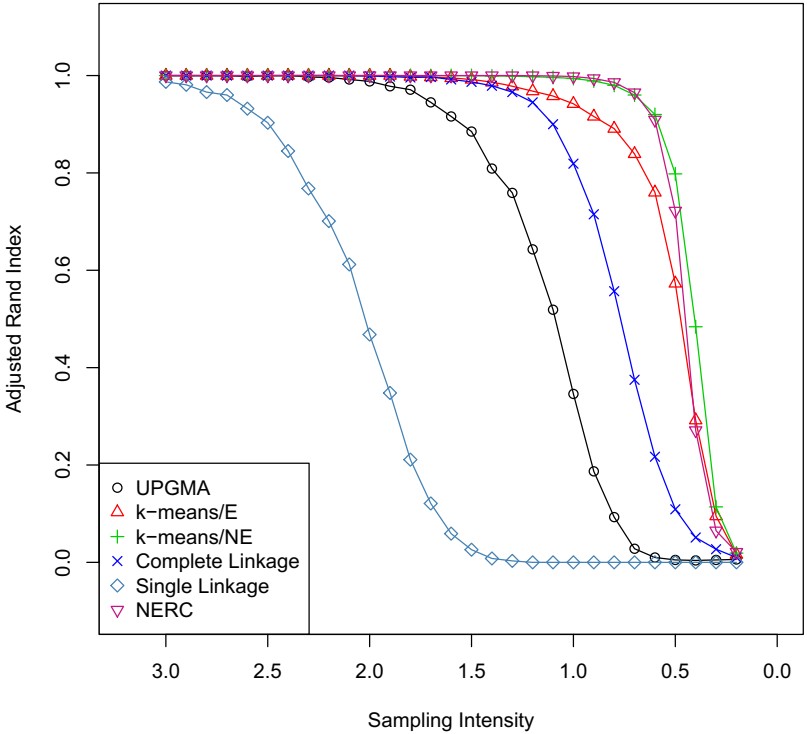

**Figure 3 Comparison of cluster methods with varying sampling intensity.** Accuracy of various clustering methods in response to changing levels of sampling intensity (coverage). Overall, as sampling intensity decreases (to the right), clustering becomes less reliable. The values for each method at any given level of sampling is the average Adjusted Rand Index comparing the known solution and the calculated solution over 1000 simulations.

the other clustering methods, single linkage performed the poorest of any of the methods. The notably poor performance of the single linkage method was likely the result of individual samples that were extremely distant from all others placed at the base of the tree, and because I applied a strict tree cutting method with the hierarchical methods to obtain discrete clusters, the tree cutting method then identified this single distant sample as an individual cluster. However, the treatment of outliers is challenging in all clustering approaches, and their exclusion may not be possible or desirable. A similar situation, where outliers have an undue influence on group composition, is likely why complete linkage and UPGMA are also less effective than *k*-means or NERC.

These hierarchical methods are well suited to applications such as phenetic analyses or phylogenetics, where a single ancestor (theoretically) gives rise to multiple descendants. However, this one-to-many structure often translates poorly to species occurrence data sets like those commonly used in biogeographic studies, where individual lineages may be operating in parallel and independently (*Brown, 1999*). Individual species may originate in different locations and disperse by various methods to new regions (*Brown, 1999*), leading to a more reticulate, many-to-many relationship. In this case, a method that does not enforce a hierarchy may better represent the relationships present.

Further, species occurrence data is typically non-Euclidean in nature. Whereas all the cells in a phylogenetic data matrix represent a directly observed value, in a species occurrence matrix any cell that has a zero value may be due to either the species not occurring in that area or incomplete sampling, two possibilities that may be indistinguishable from one another. To deal with incomplete sampling, most species occurrence data sets are converted into a distance matrix, where the species composition of each sample is compared to every other sample using an index of similarity (or dissimilarity); yet, while most of these measures provide some measure of distance, these distances are not necessarily Euclidean (*Gower & Legendre, 1986*). The benefit of using non-Euclidean measures over Euclidean distances is readily observable in this study, with the non-Euclidean based *k*-means outperforming the Euclidean based *k*-means.

Although for this study, the Sørensen dissimilarity index was used, the choice of which non-Euclidean dissimilarity index to use is not necessarily straightforward (e.g. *Shi, 1993*; *Magurran, 2004*; *Alroy, 2015*). By some counts, dozens of different dissimilarity indices have been proposed in the literature (*Hubálek, 1982*; *Pielou, 1984*; *Shi, 1993*), although only a handful of these have entered into common use (*Magurran, 2004*). While alternative methods, such as a recent modification to the Forbes metric (*Alroy, 2015*), have been proposed as replacements to more traditional dissimilarity metrics, the choice of measure is a separate question to the issue in the present study. Although using other dissimilarity measures may have changed the individual effectiveness of the different clustering methods, the relative performance of the clustering methods to each other is unlikely to change, as even with different measures the problems of outliers and hierarchical/non-hierarchical methods would persist.

Both poor differentiation between clusters and inadequate sampling are common problems with palaeobiological data. No method is entirely immune to either of these issues, but overall, based on these simulations, *k*-means and NERC give more reliable and accurate results when data are less than robust. Using these methods still does make one strong assumption about the underlying data–namely, that true divisions within the data exist. Unfortunately, with the often muddled and noisy nature of biogeographic data, this assumption is also the hardest to objectively determine.

## ACKNOWLEDGEMENTS

I would like to thank Caleb M. Brown, Nicolás E. Campione, Luke B. Harrison, Pat A. Holroyd and Brian D. Rankin for their critical feedback which improved the quality of this paper, and Callum G. Vavrek and Ada E. Vavrek for their assistance in editing the manuscript.

### Funding

This work was funded in part by a National Sciences and Engineering Research Council Canada Graduate Scholarship (Doctoral). The funders had no role in study design, data collection and analysis, decision to publish, or preparation of the manuscript.

## Competing Interests

The author declares that they have no competing interests.

## Author Contributions

- Matthew J. Vavrek conceived and designed the experiments, performed the experiments, analyzed the data, contributed reagents/materials/analysis tools, wrote the paper, prepared figures and/or tables, reviewed drafts of the paper.

## Data Deposition

All R code used in this analysis has been uploaded as a Supplemental File.

## Supplemental Information

Supplemental information for this article can be found online at http://dx.doi.org/10.7717/peerj.1720#supplemental-information.

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
