# Peer review of "A comparison of clustering methods for biogeography with fossil datasets"

_PeerJ, doi:10.7717/peerj.1720_

## Round 0.1 · original submission · Minor Revisions

The manuscript has been carefully revised by two external reviewers and it will require minor revisions to address the point raised by the reviewer #1 that are hereby attached. I am looking forward receiving the revised version of this very nice and interesting contribution.

Reviewer 1 ·

Basic reporting

The manuscript is written in clear and concise English.

The background information is sufficient to understand the problems addressed by the author.

The figures are relevant and informative.

This manuscript is certainly a "publishable unit."

Experimental design

See my comments to the author.

Validity of the findings

The findings in this paper are valid. The conclusions follow naturally from the results. See my comments to the author.

Additional comments

This paper uses simulation to assess the efficacy of various clustering methods for distinguishing biogeographic provinces. It also introduces a new clustering method (NERC). I believe the problem posed in this study is very important, as the author states, given increased data availability and efforts to understand biodiversity patterns in the past. This paper is a good guide for those interested in paleobiogeography who are unfamiliar with cluster analyses. The problem and question is clearly defined and is relevant to the burgeoning study of paleobiodiversity. I know of no other comprehensive study regarding the performance of clustering methods using data typical of the fossil record. The paper therefore fills a very clear void in the paleobiodiversity literature. I don’t have any major concerns about this manuscript and think it deserves to be published with only some minor revisions.

I have played with the functions described in this manuscript and provided in the fossil package. They seem to work well. However, I feel that this paper could benefit from providing documentation for at least the new clustering method it presents. The actual code for the NERC clustering method is not provided. I had fully expected to find at least an R markdown document describing the rclust() function. It would have been easier to evaluate the function itself if such materials had been provided.

The paper states that sim.clusters() is used. I cannot find this function in the fossil package and it is also not used in the provided R code. The author instead uses sim.occ(). I am sure this is simply a typing error or a change in the name of the function that is not yet reflected in the fossil package.

The author discusses "presence only" matrices (line 166; matrices with only 0’s for absence and 1’s for presence) but doesn’t appear to have tested the various clustering methods on them. As you reduce the avg.abund in the sim.occ() function, you can get “presence only” matrices but you’re also reducing richness at each site. The efficacy of these methods when we possess presence only data is paramount because it is my opinion that the abundance data in the various fossil databases are far too sparse. Of course, I think the results presented in this paper are probably generalizable to presence only matrices (a sampling intensity of 0.2 or lower gets pretty close and the performance of the various clustering methods remains the same according to figures 1 and 2 = k means and NERC outperforming the others). However, due to the way sim.occ() functions (high values of avg.abund mean both high richness and abundance), it is less clear how the clustering methods perform when richness is high but abundance data are missing.

Following my last comment, does choice of dissimilarity metric affect the outcome of these simulations? This might be particularly important for presence only matrices (see Jaccard vs Alroy’s modified Forbes Index; http://www.esajournals.org/doi/abs/10.1890/14-0471.1). I am not necessarily proposing that the author run a huge number of additional simulations but there is no discussion of this in the manuscript. Having run only 10 iterations using the provided code, I can see how long the simulations for this paper likely took (I did not use parallel computing, however).

I am also wondering how endemicity and sampling intensity interact in this context. I am certain that there are instances in the past when both parameters vary simultaneously. Do all of the clustering methods simply perform poorly in this case? Are we doomed never to understand biogeographic patterns in the past? I hope not! Where might the real fossil record lie on figure 3? If it’s somewhere between 3 and 0.5, we’re pretty golden when using the NERC and k means clustering methods. If it’s considerably lower, which might be the case for at least some floras and faunas, we are probably hopelessly doomed. In general, the author could discuss some of these issues at the end of the paper. It might function to tie the simulations back into real studies of the fossil record.

Minor things

In the first paragraph cite the creators of MioMAP, FAUNMAP, and the NOW database. This information should be found on their respective websites.
Line 75: optimum not optima
Line 105: A log-normal distribution is standard for simulating species abundance distributions but the author should provide citations to support it.
Line 125: Italicize a priori

Reviewer 2 ·

Basic reporting

The article is well written and succinct.

Experimental design

The experimental design is straight-forward and sufficiently interesting.

Validity of the findings

The results and discussion are clear and appropriate.

---

## Round 0.2 · accepted · Accept

All the previous concerns have been fully addressed and we are now ready to endorse this nice contribution for publication.